# Amazon Fruits Inhibit Growth and Promote Pro-apoptotic Effects on Human Ovarian Carcinoma Cell Lines

**DOI:** 10.3390/biom9110707

**Published:** 2019-11-06

**Authors:** Vanessa Rosse de Souza, Mariana Concentino Menezes Brum, Isabella dos Santos Guimarães, Paula de Freitas dos Santos, Thuane Oliveira do Amaral, Joel Pimentel Abreu, Thuane Passos, Otniel Freitas-Silva, Etel Rodrigues Pereira Gimba, Anderson Junger Teodoro

**Affiliations:** 1Laboratory of Functional Foods, Universidade Federal do Estado do Rio de Janeiro, Rio de Janeiro 22290-240, Brazil; vanessa_rosse@hotmail.com (V.R.d.S.); thuane.amarall@gmail.com (T.O.d.A.); pimenabreu@gmail.com (J.P.A.); thuanepassos@gmail.com (T.P.); 2Cellular and Molecular Oncobiology Program, Research Centre, Instituto Nacional de Câncer, Rio de Janeiro 20231-050, Brazil; mariana.concentino@gmail.com (M.C.M.B.); paulapdefreitas@gmail.com (P.d.F.d.S.); etelgimba@id.uff.br (E.R.P.G.); 3Division of Clinical Research, Research Center, Instituto Nacional de Câncer, Rio de Janeiro 20231-050, Brazil; isaguimaraes@hotmail.com; 4Embrapa Agroindústria de Alimentos, Empresa Brasileira de Pesquisa Agropecuária, Rio de Janeiro 23020-470, Brazil; otniel.freitas@embrapa.br; 5Department of Nature Sciences, Universidade Federal Fluminense, Rio das Ostras 28895-532, Brazil

**Keywords:** *Byrsonima crassifolia*, *Byrsonima verbascifolia*, *Spondias mombin*, carotenoids, cisplatin, ovarian cancer

## Abstract

Murici (*Byrsonima crassifolia* (L.) Kunth and *B. verbascifolia* (L.) DC.) and tapereba (*Spondias mombin*) are Amazonian fruits that contain bioactive compounds. Biochemical and molecular characterization of these fruits can reveal their potential use in preventing diseases, including cancer. The extracts were characterized regarding the presence and profile of carotenoids by high-performance liquid chromatography (HPLC), total phenolic content by the Folin–Ciocalteu assay, and antioxidant activity by antioxidant value 2,2-diphenyl-1-picrylhydrazyl (DPPH) content analysis, 22,20-azino-bis(3-ethylbenzothiazoline-6-sulfonic acid) (ABTS) content analysis, Ferric-Reducing Ability of Plasma (FRAP), and Oxygen Radical Absorbance Capacity (ORAC) analysis. The extracts of tapereba and murici studied were important sources of total carotenoids and lutein, respectively. The extracts were then tested for their effect on the viability of the A2780 ovarian cancer (OC) cell line and its cisplatin (CDDP)-resistant derived cell line, called ACRP, by using MTT (3-(4,5-dimethylthiazol-2-yl)-2,5-diphenyltetrazolium bromide) assays. Their influence on cell cycle and apoptosis were analyzed by using flow cytometry. Murici and tapereba cell extracts exhibited a strong bioactivity by inhibiting A2780 and ACRP cell viability by 76.37% and 78.37%, respectively, besides modulating the cell cycle and inducing apoptotic cell death. Our results open new perspectives for the development of innovative therapeutic strategies using these Amazon fruit extracts to sensitize ovarian cancer cells to current chemotherapeutic options.

## 1. Introduction

Ovarian carcinoma (OC) is a highly heterogeneous disease and despite advances in understanding it, patients suffering from this disease still have poor prognostic rates related to late diagnosis and rapid progression [1]. In addition, the disease has the ability to metastasize into the peritoneal cavity [2]. Its main treatment is cytoreductive surgery followed by platinum/taxane-based chemotherapy in advanced cases. Cisplatin (CDDP) is considered the first line of treatment against OC and despite high initial response rates, a large percentage of patients relapse after treatment. Furthermore, high doses of CDDP are related to irreversible side effects and resistance [3].

Alternative approaches or combination therapies including CDDP could be important options for the treatment of OC. It is known that Brazil has the greatest biodiversity in the world, with many species still unknown or little studied. Fruit species have high economic values, both in the fresh fruit trade and use as raw materials for agroindustry [4]. The Amazon region contains a wide range of fruit varieties with economic potential due to their diverse aromas and exotic flavors [5].

Among the main preventive strategies for health is chemoprevention, which consists of the use of natural or synthetic chemical agents to prevent, interrupt, stabilize, or reverse the genesis of cancer [6,7]. Some of the priorities of this research area are the use of phytochemicals and chemopreventive compounds in fruits, vegetables, and other plants, which have been proposed as auxiliary tools for cancer prevention [8,9]. The Amazon region contains a vast diversity of natural products to be explored. Several studies have found that the fruits murici (ME) and tapereba (TAP) are sources of bioactive compounds with antioxidant activity, indicating their importance to human health [10,11,12,13].

These fruits are endemic in some ecosystems in Brazil, but produced predominantly in the Amazon region in forest yards and are then harvested by extractivism. They present an increasing economic importance in tropical regions, especially in the field of eccentric juices and desert markets [12,13]. The yield of these fruits according to the Brazilian yearbook of geography and statistics is about 7.8 tons/year in 2018. This activity contributes greatly to the increase in family farmers’ incomes and plays a key role in the local economy by accounting for more than $2 million/year [14].

Impressive progress is being made in discovering the role of bioactive compounds in reducing the risk of cancer and the underlying biological mechanisms that account for these effects. Epidemiological studies suggest that cancer risk is related to dietary intake of fruits and vegetables that are rich in carotenoids and phenolic compounds [9]. These bioactive compounds have been linked to many health benefits, including cancer prevention, for which different action mechanisms have been identified. In general terms, under oxidative stress, polyphenols and carotenoids can act in those cellular mechanisms by helping to modulate the redox status and affect multiple key elements in intracellular signal transduction pathways related to cell proliferation, differentiation, apoptosis, inflammation, angiogenesis, and metastasis [9].

ME has been described as a good source of lutein and zeaxanthin [10]. Besides these carotenoids, a series of phenolic compounds derived from quercetin and galloyl have also been identified [11] that give this fruit a high antioxidant activity [12]. Notably, ME extracts contain carotenoids such as zeaxanthins, β-cryptoxanthin, and α-carotene [13]; phenolic compounds; saponins; tannins; flavonoids; alkaloids; and glycosides [15].

TAP is also a source of phenolic compounds having tannins as the main component. Its antioxidant capacity may also be related to the presence of compounds such as vitamin C and carotenoids. Large amounts of gallic acid and quercetin have been found in TAP, and its pulp can effectively inhibit oxidation, which is attributed to its yellow flavonoids, carotenoids, and chlorophyll [13,16,17].

Based on the need to develop more effective treatment strategies to reverse chemoresistance and the survival rates in OC patients and the potential properties of ME and TAP as sources of bioactive compounds, this paper investigates the use of extracts from these Amazonian fruits as auxiliary tools to modulate the cell viability and survival of A2780 and ACRP OC cell lines.

## 2. Materials and Methods

### 2.1. Samples

Pulps of ME and TAP packaged in sealed and labeled plastic bags (1 kg) were supplied by a company from Pará (PF, Castanhal, PA, Brazil) and stored at a controlled temperature (−18 °C). The frozen pulp was transported in an ice chest containing dry ice to the Laboratory for Analysis of Functional Foods (LAAF-UNIRIO), Rio de Janeiro (Brazil), where they remained frozen (−18 °C) until the moment of analysis.

### 2.2. Extraction of Samples

Approximately 250 g of pulp of ME and TAP was extracted with 80 mL of distilled water and then shaken for 2 h. After the pulp maceration period, the aqueous ME and TAP extracts were filtered through Whatman #1 filter paper. The extracts were then frozen at −80 °C in an ultra-freezer and lyophilized (Terroni^®^ LD 300, São Carlos, SP, Brazil) for 24 h. After this process, extracts were frozen at −20 °C until use in the experiments [18].

### 2.3. Carotenoid Analysis

The composition and level of carotenoids in extracts of powdered ME and TAP pulps were determined using high-performance liquid chromatography (HPLC, W600 - Waters^®^, Milford, KS, USA) according to Rodriguez-Amaya (2001) [19]. Saponification was performed to remove interfering substances and the saponified extract was filtered and analyzed by spectrophotometry to quantify the total carotenoid content. After quantification, the extracts were concentrated to carry out the separation. The separation was performed with a C30 column (YMC Carotenoid 3 µm (4.6 × 250 mm), Waters^®^, Milford, KS, USA), with 80% MeOH (Tedia, Fairfield, OH, USA), and 20% methyl t-butyl ether as the mobile phase with a column temperature of 33 °C. β-Cryptoxanthin, lutein, zeinoxanthin, α-carotene, β-carotene, and zeaxanthin were quantified.

### 2.4. Determination of total Phenolics

The Folin–Ciocalteu assay was performed to determine the concentration of total phenolics in the ME and TAP aqueous extract powders. The method was performed as described by Singleton and Rossi (1999) [20]. The extracts were added to 2.5 mL of Folin–Ciocalteu reagent and 2 mL of 4% sodium carbonate solution and the mixture was allowed to rest for 2 h in the dark. A standard gallic acid curve was used. Absorbance was read at 750 nm by spectrophotometry (Turner^®^ 340, Haverhill, MA, USA) in triplicate and the results were expressed as mg gallic acid equivalent (GAE)/mL extract.

### 2.5. Antioxidant Activity Analysis

#### 2.5.1. Oxygen Radical Absorbance Capacity (ORAC) Assay

The antioxidant capacity of the ME and TAP extracts was measured by the ORAC assay, according to Prior and Hoang (2000) [21]. PBS (pH 7.4), a fluorescein solution, Trolox standard, and a 2,2’-azobis(2-amidinopropane) dihydrochloride (AAPH) solution were prepared for this purpose. The Trolox standard was prepared at eight different concentrations (2.5 to 20 μg/mL). For blank aliquots and control, the phosphate-buffered saline (PBS) solution was used. The Trolox standard and extracts were added to the plate in increasing concentrations and in duplicate. Then 120 μL of the fluorescein solution was added to all wells followed by 60 μL of the AAPH solution, except for the control. The fluorescence drop reading was measured using an automated plate reader (SpectraMax i3x, Molecular Devices, USA) with 96-well plates at 485/520nm (excitation/emission). The calculation was performed considering the area under the curve (AUC).

#### 2.5.2. Ferric-Reducing Ability of Plasma (FRAP) Assay

The determination of the antioxidant activity of the ME and TAP extracts through the FRAP assay was performed according to the method described by Rufino et al. (2006) [22], by the reaction of 0.5 mL of ME and TAP extracts with 2.7 mL of ferric acid with 2,4,6-tripyridyl-s-triazine reagent (TPTZ). After 30 min at 37 °C, the absorbance was read at 595 nm and the results were expressed as μmol ferrous sulphate/g.

#### 2.5.3. Scavenger Radical Activity (DPPH)

A reaction mixture of 2.5 mL DPPH methanolic solution (0.06 mM) with 0.5 mL of ME or TAP extracts was mixed completely and left in the dark at a controlled temperature for 60 min. The absorbance of the blend was measured with a spectrophotometer (Turner 340) at 515 nm in triplicate [23].

#### 2.5.4. Trolox Equivalent Antioxidant Capacity (TEAC) Assay

TEAC assay was performed following the procedure proposed by Rufino et al. (2007) [24]. The ABTS radical (7 mM) was prepared and kept in the dark at room temperature for 16 h before use. The ABTS solution was diluted with ethanol AP to attain absorbance of 0.70 ± 0.02 at 734 nm. After adding 30 μL of ME or TAP extract or Trolox standard (five concentrations) to 3 mL of diluted ABTS solution, the absorbance was recorded six minutes after addition. The analyses were performed in triplicate with a spectrophotometer (Turner 340). The blank assay used ethanol and antiradical activity was expressed as μmol TE/g.

### 2.6. Cell Culture and Treatment Protocol

The human ovarian carcinoma cell line (A2780) was obtained from the National Cancer Institute (INCA, Rio de Janeiro, RJ, Brazil) and the CDDP-resistant cell line (ACRP) was developed using a pulse selection strategy followed by recovery in free media of drugs to mimic the clinical administration of chemotherapy. The two cell lines (A2780 and ACRP) were routinely maintained in an RPMI-1640 medium (Gibco, Gaithersburg, MD, USA) supplemented with 10% fetal bovine serum (FBS) (Gibco, Gaithersburg, MD, USA) and 1% penicillin, under 5% CO_2_ atmosphere. Stock cultures in flasks were grown to 80% confluence and routinely subcultured. Cell morphology was observed using a Zeiss Observer Z1 microscope and all images were captured using Axio-Vision Rel. 4.8 software (Carl Zeiss, Jena, Germany). For each experiment, the cells were seeded at 2 × 10^5^ cells/cm^2^ and 5 × 10^5^ cells/cm^2^ densities in 96-well plates and 6-well plates.

### 2.7. Cytotoxic Analysis

#### 2.7.1. MTT Assay

The cytotoxic effects of ME and TAP extracts were monitored by the 3-(4,5-dimethylthiazol-2-yl)-2,5-diphenyltetrazolium bromide (MTT) assay. A2780 and ACRP cells were seeded at 2 × 10^5^ in 96-well plates in triplicate and incubated for 24 h following the procedure for cell adhesion. Then the medium was removed and the cells were exposed to nine concentrations of both the extracts (0.01 to 20 mg/mL) for 24 h. The treatment with fruit extracts in combination with CDDP, previously defined at 10 µM and 80 µM, was set for the A2780 and ACRP cell lines, respectively [25], while for the extracts, four different concentrations (5 to 20 mg/mL) were used. The untreated medium was added to the control wells. Upon completion of the exposure, 20 μL of MTT (5 mg MTT/mL) was added to each well. After four hours, the MTT solution was removed and the insoluble formazan crystals were dissolved in 150 μL of DMSO (Sigma-Aldrich, St. Louis, MO, USA). Optical density was determined using a Flex Station 3 (Molecular Devices Corporation, St. José, CA, USA). All experiments were repeated three times.

#### 2.7.2. Cell Cycle

Human ovarian cancer cell lines (A2780 and ACRP) were seeded at 5 × 10^5^ cells/cm^2^ in a 6-well plate. Cells received treatment with ME and TAP extracts at two concentrations (5 and 20 mg/mL). After incubation for 24 h, the cells were briefly washed with PBS solution and resuspended in 500 μL of Vindelov’s solution containing 0.1% Triton X-100, 0.1% citrate buffer, 0.1 mg/mL RNAse, and 50 mg/mL propionate iodide (Sigma Chemical Co., St. Louis, MO, USA) [26] and left for 15 min at room temperature. Doublets and debris were identified and excluded. Approximately 30,000 cells were used for each analysis, samples were run at a low flow rate and the distribution of cells in each phase of the cell cycle was displayed as histograms. The quality of cell cycle data was assessed using the coefficient of variation (CV) of the G1 peak. A CV of <6% was considered acceptable. Stained cells were detected with BD Accuri ™ C6 flow cytometer and quantified with the CFlow^®^ software (BD Accuri^TM^, Franklin Lakes, NJ, USA).

#### 2.7.3. Apoptosis Assays

The ovarian cancer cell lines were seeded in 6-well plates at concentrations similar to those used for cell cycle analysis. After 24 h of treatment with the extracts at concentrations of 5 and 20 mg/mL, the cells were washed with buffered saline solution (PBS), resuspended in a binding buffer with 5 μL of annexin V FITC and 5 μL of propidium iodide (PI) (Apoptosis Detection Kit II, BD Pharmingen, New Jersey, USA). Apoptotic and necrotic cells were labeled and quantified in a BD Accuri ™ C6 flow cytometer and analyzed with the CFlow^®^ software (BD Accuri^TM^, Franklin Lakes, NJ, USA).

### 2.8. Statistical Analysis

The results presented are the mean and corresponding standard deviations of three independent experiments performed in triplicate (*n* = 9). Data were analyzed using GraphPad Prism (version 5.04, GraphPad Software, San Diego, CA, USA). A univariate analysis of variance (ANOVA) and the Tukey’s post-test at a 95% confidence level were used to test cell viability, cell cycle, and apoptosis rates.

## 3. Results and Discussion

### 3.1. Bioactive Properties of Murici (ME) and Tapereba (TAP) Extracts

ME and TAP extracts presented high antioxidant activity and were able to confer benefits to human health. ME presented the highest antioxidant activity in the assays for oxygen radical absorbance capacity (ORAC) (1020.39 ± 88.43 μM TE/g), ferric-reducing ability of plasma (FRAP) (1014.71 ± 2.08 μmol ferrous sulfate/g), and Trolox equivalent antioxidant capacity (TEAC) (1620.95 ± 114.65 μmol TE/g), when compared with TAP, which presented mean values of 623.72 ± 38.75 μM TE/g (ORAC), 644.55 ± 10.89 μmol ferrous sulfate/g (FRAP), 78.70 ± 0.28% reduction in (2,2-diphenyl-1-picrylhydrazyl hydrate) free radicals (DPPH), and 1090.90 ± 296.04 μmol TE/g (TEAC). On the other hand, TAP presented a higher reduction activity in the DPPH assay compared with the ME extract (Table 1). These values are similar to those found by Tiburski et al. [13] in their study of TAP pulp.

ME contained higher levels of total phenolic compounds (1634.05 ± 278.18 mg gallic acid (GAE)/mL) compared with TAP (1049.09 ± 95.68 mg GAE/mL), which might explain the antioxidant potential of this extract. Almeida et al. [27] studied fresh exotic fruits from northeastern Brazil and observed results of phenolics content (1599.0 ± 56.00 18 mg GAE/g) similar to our findings in ME pulps.

In other studies, approximately 19 polyphenolic compounds have been identified in murici, including galotannins, quinic acid gallates, proanthocyanidins, quercetin derivatives, and gallium derivatives—substances rarely found in fruits [10,28]. Authors have found strong correlations between total phenolic compounds and antioxidant activity in fruits, including murici [27]. Researchers have already identified saponins, tannins, flavones, flavonoids, leucoanthocyanidins, alkaloids, and glycosides in tapereba pulps [15].

TAP presented higher levels of total carotenoids (185.92 ± 12.86 μg/g) compared with ME (86.30 ± 8.82 μg/g). Matietto et al. [17] found total carotenoid values for TAP pulps ranging from 10.71 to 37.55 μg/g and 30.30 to 38.56 μg/g. According to Rodriguez-Amaya et al. [28], changes in the carotenoid content of the same food are possible due to growing conditions in Brazil.

In the present study, six carotenoids were identified in both the pulps: β-cryptoxanthin, lutein, zeinoxanthin, α- and β-carotene, and zeaxanthin. Among these carotenoids, β-cryptoxanthin, α-carotene, and β-carotene have pro-vitamin A activity [29]. β-cryptoxanthin (89.81 ± 4.58 μg/g) and lutein (23.39 ± 1.41 μg/g) were the major components among the carotenoids identified in TAP and ME, respectively (Table 2). Lower levels were found for zeinoxanthin, α-carotene, and β-cryptoxanthin in ME. Hamacek et al. [30] identified only β-carotene, not α-carotene, and β-cryptoxanthin in ME pulp samples, unlike our results.

Carotenoids are a group of pigments widely distributed in nature that occur in large quantities. These compounds are known for their structural diversity and various biological functions [31]. In fruits, the carotenoid content increases with the maturation process, and part of their color intensification is due to the degradation of chlorophyll [32].

Tiburshi et al. [13] found higher levels of carotenoids (β-cryptoxanthin, α-carotene, and β-carotene) in the TAP pulp than the levels found in the present study. The Amazon has many native fruits that are good sources of carotenoids such as buriti, mamey, marimari, pupunha, physalis, tucuma, and apricots [27,33]. Braga et al. [5] evaluated the levels of carotenoids in apricot pulp powders and found lower levels of carotenoids compared with the levels found in the present study [27]. Rosso and Mercadante [34] verified that the total carotenoid content varied from 38 μg/g in marimari to 514 μg/g in buriti [33]. Our data highlight the strong potential of ME and TAP as sources of carotenoids—bioactive compounds that are widely present in Amazonian fruits.

### 3.2. Cytotoxic Analysis—MTT

In order to evaluate the effect of ME and TAP on the viability of the two ovarian cancer cell lines, A2780 and its CDDP-resistant derivative ACRP, these were treated with 0.001–20 mg/mL of each extract for 24 h. ME at a concentration of 20 mg/mL promoted 77.38% and 83.94% decreases in A2780 and ACRP cell viability, respectively, compared with the control group (*p* < 0.05) (Figure 1). At the same concentration (20 mg/mL), TAP promoted a significant reduction (*p* < 0.05) in A2780 (69.40%) and ACRP (65.54%) cell viability (Figure 2). MTT tests using 5 or 10 mg of each extract presented a similar cell viability reduction in both the extracts (*p* > 0.05). When tested at a concentration of 5 mg/mL, TAP promoted a mean reduction of 58.22% in A2780 cell viability. However, it did not promote reduction in A2780 cell viability at the concentrations of 5 and 10 mg/mL compared to ACRP cells, as seen in Figure 1 and Figure 2.

Pessoa et al. [35] in their review study reported the results of experiments with ME and TAP, also demonstrating that ME (75.0 mg) promoted a reduction of 26% in cell viability. Regarding the treatment of the metastatic tumor cell line with 125 mg of TAP, we found an 18% decrease in cell viability. Thus, our data showed that both ME and TAP extracts at lower concentrations than those used in previous studies had higher efficacy regarding A2780 and ACRP ovarian cancer cell viability.

ME promoted a greater reduction of cell viability in relation to the control than the TAP extract. This decline was also higher in the parental ovarian cancer cell line (A2780) than in the ACRP resistant cell line (*p* < 0.05).

### 3.3. Effect of ME and TAP on Cell Cycle Progression

Since ME and TAP promoted a significant reduction of A2780 and ACRP cell viability, we then investigated whether this effect could be due to cell cycle modulation. Cell cycle analyses were performed using 5 mg/mL of both ME and TAP, since with a concentration of 20 mg/mL, it was impossible to fully visualize the cell cycle due to the high rate of cell death.

Percentage increases were found of cells in the G_0_/G_1_ phase after treatment with ME (54.56 ± 1.55%) and TAP (68.80 ± 5.82%) in A2780 parental cells. The same effect was observed for ME (19.03 ± 3.09%) and TAP (36.60 ± 11.03%) in ACRP, the CDDP-resistant cell line. A decrease of cells in the S and G_2_/M phases in response to cell treatment with both the extracts was also observed (Figure 3A,B), this effect being greater in ACRP than the A2780 parental cell line (*p* < 0.05). The highest concentration of cells in the G_0_/G_1_ phase was observed in response to cell treatment with ME in the two cell lines studied (*p* < 0.05), indicating cell growth arrest of A2780 and ACRP after treatment (Figure 3A).

A deregulated cell cycle is one of the main characteristics of cancer, related to initiation and progression. Thus, its regulation is of extreme importance to control the disease, making its study important for the development of cancer treatment [36,37]. A compound or extract can be considered efficient in the treatment of cancer when it is able to block the initiation and propagation phases of the cell cycle, i.e., the G_0_/G_1_ and G_2_/M phases, thus reducing the number of cells in the S phase [38].

Studies have demonstrated the influence of carotenoids on cycle arrest [39]. Some specific carotenoids, like lutein and ß-carotene, have been shown to inhibit proliferation in the G_0_/G_1_ phase [40]. ME and TAP fruits are important sources of these carotenoids, which gives them a high antioxidant activity and the ability to modulate the cell cycle. No studies have been published evaluating the cell cycle modulation in ovary cancer cell lines using natural compounds.

### 3.4. Effect of Murici and Tapereba Extracts in an Apoptosis Assay

An uncontrolled increase in cell proliferation and a reduction of apoptosis have been associated with cancer metabolism. In the present study, we evaluated the apoptotic rates after 24 h of treatment with ME and TAP at concentrations of 5 and 20 mg/mL. The extracts caused an increase in the percentage of apoptotic cells in relation to untreated cells in the A2780 parental cell line. In this cell line, ME at 5 mg/mL and 20 mg/mL respectively, promoted increases of 38.54 ± 7.84% and 45.11 ± 1.28% of the apoptotic rates. TAP at 5 and 20 mg/mL, respectively, promoted increases of 46.51 ± 0.13% and 46.50 ± 0.28% in the apoptotic rates. No significant differences were observed in the apoptotic rates between the two tested ME concentrations (Figure 4, top panel).

However, when ACRP cells were treated under the same conditions for 24 h with TAP, no effects on the number of apoptotic cells were detected, evidencing this cell line has a different behavior in relation to parental cells, possibly related to its CDDP-related resistance. Also, another cell death pathway could be involved in the decreased viability in addition to inhibition of cell cycle progression (Figure 4).

Phenolic compounds and carotenoids have been indicated for their ability to reduce the incidence of cancer in humans through the activation of cell apoptosis and the production of reactive oxygen species. The parental ovarian cancer cell line presented elevated cell death by apoptosis after treatment with ME and TAP extracts, which are sources of these compounds, suggesting their action in the regulation of apoptosis and control of the disease. Cui et al. [41], in a study with breast cancer cells using β-carotene, demonstrated apoptosis regulation after this treatment, similar to the present study. The increase noted in the present study was probably higher due to the interaction of the compounds present in the fruits. No studies similar to the one performed in ovarian cancer cells have been published.

### 3.5. Cytotoxic Analysis—MTT after Combined Treatment

A previous in vitro study performed with human ovarian parental cancer cells (A2780) and resistant cells (ACRP) demonstrated that the effective dose of CDDP treatment in these situations ranges from 10 µM for A2780 to 80 µM for ACRP cell lines [19]. Based on these results, we examined the combined treatment, considering the CDDP doses of 10 µM and 80 µM for A2780 and ACRP, respectively, and extract doses of 5–20 mg/mL.

A2780 cells showed a greater reduction in cell viability after 24 h of treatment with ME and TAP extracts at the concentration of 20 mg/mL. A2780 cells incubated with the lowest concentration of the ME extract (5 mg/mL), as well as those treated with 10 μM of CDDP and a combination of CDDP10 and ME5, presented a similar reduction in cell viability. The treatments combining CDDP10 and TAP extract at the two selected concentrations (5 and 20 mg/mL) showed a similar reduction (Figure 5).

Regarding the ACRP cell line, the treatments with the ME and TAP extracts at 20 mg/mL and the treatment combining CDDP80 plus extracts of 20 mg/mL were similar and presented a greater reduction. The treatments with both the extracts at the concentration of 5 mg/mL and combined treatment with CDDP80 and 5 mg/mL ME and TAP extracts were similar to the chemotherapy treatment tested at the concentration of 80 µM. This suggests a preponderant effect of ME and TAP extracts compared to the use of chemotherapy or their concomitant use as a way to promote better results from the synergistic effect of the compounds, especially in cases of resistance to chemotherapy. Similar studies considering fruits from the Amazon region and combined treatment with chemotherapeutic agents in order to inhibit cell viability were not found in the literature.

## 4. Conclusions

The results indicate that the important bioactive potential of ME and TAP fruits is related to their carotenoid and phenolic content. The levels were higher in the murici extracts, which presented greater action in the cell lines studied, especially in resistance to chemotherapy.

From the results obtained, we can conclude that the two Amazonian fruits, ME and TAP, have cytotoxic effects on ovarian cancer cells, leading to cell cycle arrest and apoptosis of diseased cells. The effects of these extracts may open perspectives to design combined approaches to sensitize ovarian cancer cells to current chemotherapeutic drugs such as cisplatin. Therefore, further studies are needed to clarify the putative therapeutic potential of these fruit extracts in ovarian cancer cells.

To date, studies reported in the literature about ME and TAP fruit extracts are still scarce, especially of their influence on cancer cell lines. Our study is the first to provide experimental evidence that ME and TAP fruit extracts can inhibit cell viability of parental and CDDP-resistant ovarian cancer cells. The reported studies with these fruit extracts on other cell lines have demonstrated that they are potent antioxidants, providing early evidence that can be used for the development of new chemotherapeutic strategies aiming to prevent the development of several diseases, including cancer.

## Figures and Tables

**Figure 1 biomolecules-09-00707-f001:**
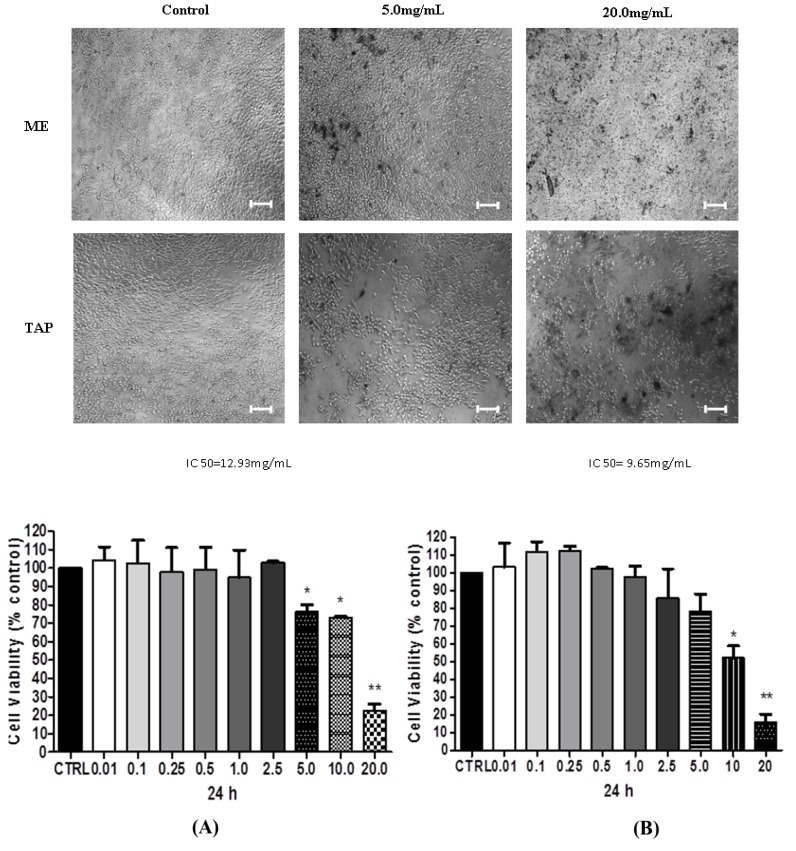
Effect of ME and TAP on A2780 cell viability. ME (**A**) and TAP (**B**) were tested for their effect on A2780 cell viability after 24 h of treatment using MTT assays. Significant differences between the untreated cells and those incubated with the respective extracts (0.01–20 mg/mL) were compared by one-way ANOVA, followed by Tukey’s post-test (* *p* < 0.05; ** *p* < 0.01). Bar 100 µm.

**Figure 2 biomolecules-09-00707-f002:**
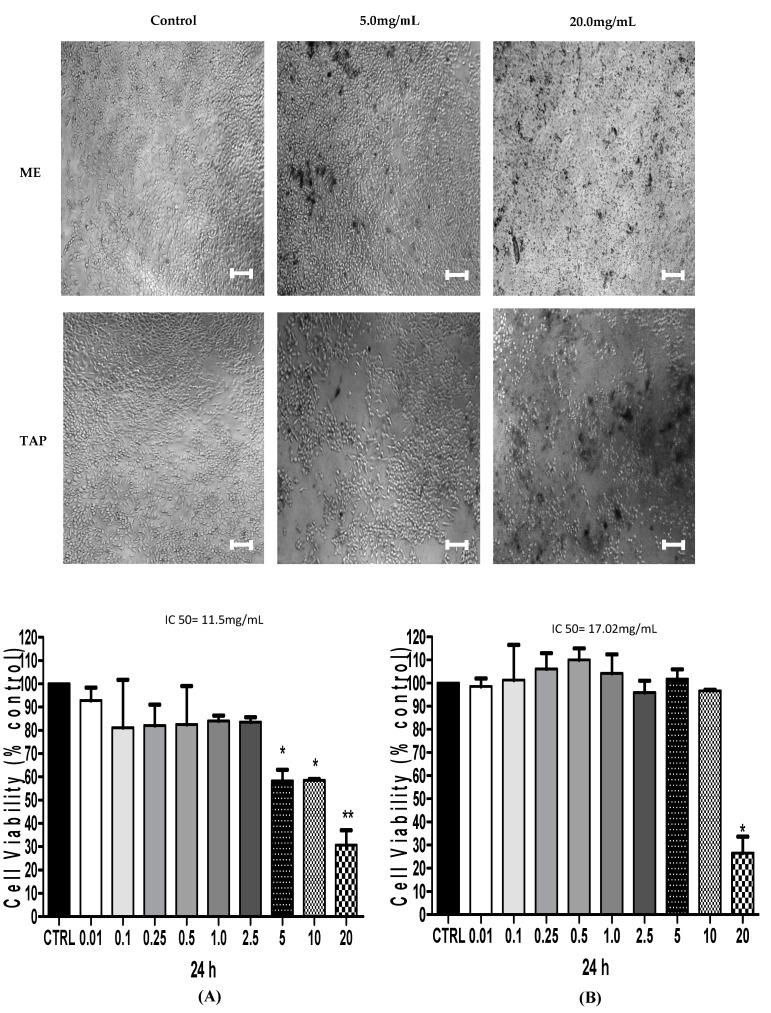
Effect of ME and TAP on ACRP cell viability. ME (**A**) and TAP (**B**) were tested for their effect on ACRP cell viability after 24 h of treatment using MTT assays. Significant differences between the untreated cells and those incubated with the respective extracts (0.01–20.0 mg/mL) were compared by one-way ANOVA followed by Tukey’s post-test (* *p* < 0.05; ** *p* < 0.01). Bar 100 µm.

**Figure 3 biomolecules-09-00707-f003:**
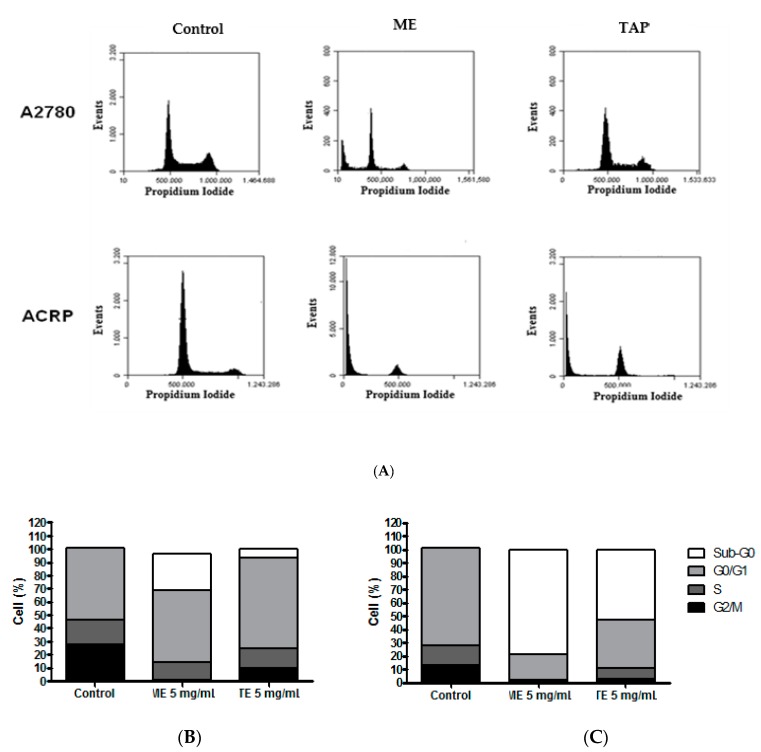
ME and TAP inhibit A2780 and ACRP cell cycle progression. A2780 and ACRP cells were tested for cell cycle progression in response to 24 h treatment with ME and TAP extracts (**A**). Flow cytometric analysis results are shown after treatment for 24 h with both cell extracts (5 mg/mL) and bar graphs represent the percentage of A2780 (**B**) or ACRP (**C**) cells in each cell cycle phase. The results are expressed as % of cells in sub-G_0_, G_0_/G_1_, S, and G_2_/M phases after cell treatment with ME or TAP.

**Figure 4 biomolecules-09-00707-f004:**
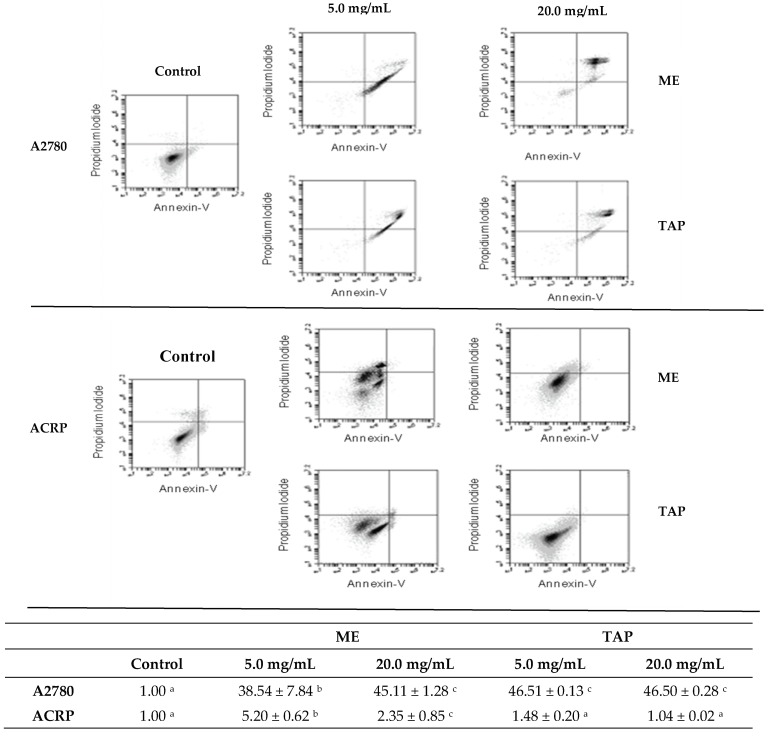
Effect of murici (ME) and tapereba (TAP) extracts on apoptotic rates in A2780 and ACRP cell lines. A2780 (top panel) and ACRP (bottom panel) cell lines were treated with ME or TAP for 24 h at 5 and 20 mg/mL. The table at the bottom part of this figure shows the apoptotic rates as described in the Material and Methods section. Different letters (a,b,c) in the same row indicate statistically significant differences (*p* < 0.05).

**Figure 5 biomolecules-09-00707-f005:**
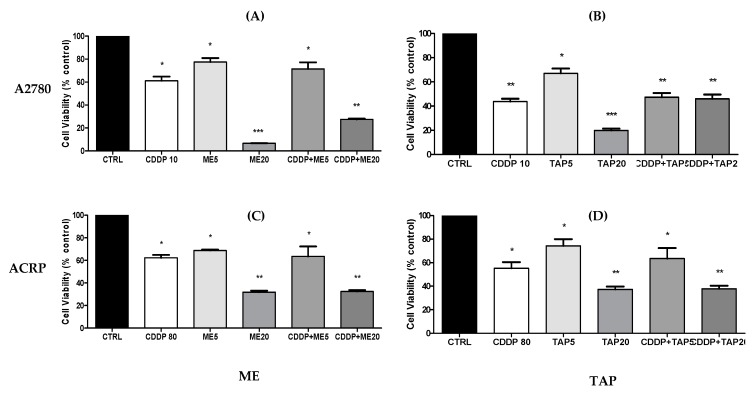
Effect of ME and TAP on CDDP of A2780 and ACRP cell viability. Effect of ME (**A**,**C**) and TAP (**B**,**D**) at the concentrations of 5 (ME5 or TAP5) and 20 mg/mL (ME20 or TAP20) on cell viability of A2780 (**A**,**B**) or ACRP (**C**,**D**) after 24 h of isolated or combined treatment with CDDP at 10 µM for A2780 (CDDP10) or 80 µM for ACRP (CDDP80) using MTT assays. The results are expressed as mean ± standard deviation of three independent experiments. Significant differences between the untreated cells (CT) and those incubated with the respective extracts were compared by one-way ANOVA followed by the Tukey’s post-test (* *p* < 0.05; ** *p* < 0.01; *** *p* < 0.001).

**Table 1 biomolecules-09-00707-t001:** Total phenolic content and antioxidant potential of ME and TAP evaluated by different assays.

Parameter	ME	TAP
Total phenolics (mg gallic acid (GAE)/mL)	1634.05 ± 278.18 ^a^	1049.09 ± 95.68 ^b^
ORAC assay (μM TE/g)	1020.39 ± 88.43 ^a^	623.72 ± 38.75 ^b^
FRAP assay (μmol ferrous sulphate/g)	1014.71 ± 2.08 ^a^	644.55 ± 10.89 ^b^
DPPH assay (% reduction)	70.17 ± 4.61 ^a^	78.70 ± 0.28 ^b^
TEAC assay (μmol TE/g)	1620.95 ± 114.65 ^a^	1090.90 ± 296.04 ^b^

Results expressed as mean ± standard deviation. Different letters (a,b) in the same row indicate significant difference (*p* < 0.05). ME = murici extract; TAP = tapereba extract; TE = Trolox equivalent; GAE = gallic acid equivalent.

**Table 2 biomolecules-09-00707-t002:** Carotenoid content of pulp extracts of murici and tapereba (µg/g).

Parameter	ME	TAP
Total carotenoids	86.30 ± 8.82 ^b^	185.92 ± 12.86 ^a^
Lutein	23.39 ± 1.41 ^a^	11.96 ± 0.07 ^b^
Zeaxanthin	5.20 ± 1.02 ^a^	1.25 ± 0.10 ^b^
Zeinoxanthin	1.92 ± 0.21 ^b^	45.72 ± 2.92 ^a^
β-cryptoxanthin	1.32 ± 0.34 ^b^	89.81 ± 4.58 ^a^
α-carotene	0.48 ± 0.11 ^b^	18.25 ± 2.99 ^a^
β-carotene	4.61 ± 1.62 ^b^	17.45 ± 3.57 ^a^

Results expressed in mean ± standard deviation. Different letters (a,b) in the same row indicate significant difference (*p* < 0.05). ME = murici extract and TAP = tapereba extract.

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
