# Peer review of "Amazon Fruits Inhibit Growth and Promote Pro-apoptotic Effects on Human Ovarian Carcinoma Cell Lines"

_biomolecules, 2019, doi:10.3390/biom9110707_

Round 1

Reviewer 1 Report

The manuscript “Amazon fruits inhibit growth and promote proapoptotic effects on human ovarian carcinoma cell lines” by Vanessa Rosse de Souza and co-authors demonstrate extracts of taperba and murici is an important source of total carotenoids and lutein, respectively. Murici and taperba cell extracts inhibited A2780 and ACRP cells viability, modulated the cell cycle and induced apoptotic cell death. The authors suggest these results might open new perspectives on the development of new therapeutic strategies using these amazon fruit extracts to sensitize ovarian cancer cells to current chemotherapeutic options. Unfortunately, the experimental design was not well organized, which significantly decreases the quality and importance of the manuscript. Also, there are other important aspects which must be taken into account before the work can be reconsidered for publication.

Comments

Figure 1: the scale bar should to be added. Figure 2: the scale bar should to be added. Figure 3: can you please provide the raw data? Figure 4: can you please provide the raw data? Can you please provide the IC50 values of fruit extracts in A2780 and ACRP cells? What is the mechanism underlying the regulation of cell cycle and apoptosis by fruit extracts? Figure 5: can you please provide the combination index in these cells? How about the cytotoxicity of the fruit extracts on normal ovarian cells?

Author Response

Reviewer 1

The manuscript “Amazon fruits inhibit growth and promote proapoptotic effects on human ovarian carcinoma cell lines” by Vanessa Rosse de Souza and co-authors demonstrate extracts of tapereba and murici is an important source of total carotenoids and lutein, respectively. Murici and taperba cell extracts inhibited A2780 and ACRP cells viability, modulated the cell cycle and induced apoptotic cell death. The authors suggest these results might open new perspectives on the development of new therapeutic strategies using these amazon fruit extracts to sensitize ovarian cancer cells to current chemotherapeutic options.

Unfortunately, the experimental design was not well organized, which significantly decreases the quality and importance of the manuscript. Also, there are other important aspects which must be taken into account before the work can be reconsidered for publication.

Comments

1 - Figure 1: the scale bar should to be added.

We appreciated this suggestion and accordingly we modified Figures 1 and 2.

2-  Figure 2: the scale bar should to be added.

As answered at # Point 1, scale bar has been also added to Figure 2.

3- Figure 3: can you please provide the raw data

We are sending as attached file all the raw data corresponding to Figure 3.

4 - Figure 4: can you please provide the raw data?

We are send also, as attached file the raw data corresponding to Figure 4. The data presented in the file corresponding to cell cycle analysis and also to the apoptosis assays.

5 - Can you please provide the IC50 values of fruit extracts in A2780 and ACRP cells?

The IC50 vales have been added to Figures 1 and 2, above the bar graphs.

6 -What is the mechanism underlying the regulation of cell cycle and apoptosis by fruit extracts?

We thank the reviewer regarding this point. Although we observed that ME and TAP extracts induced an inhibition on cell viability and also on cell cycle progression, besides promoting apoptotic cell death. But, based on our current data, we cannot establish which mechanisms of these fruit extracts can affect these processes. Future studies will be conducted by our group investigating the corresponding mechanisms and signaling the mediation by these fruit extracts.

7 -Figure 5: can you please provide the combination index in these cells?

Unfortunantelly this combination index cannot be performed using the currently available data, once we have not tested ME and TAP extracts in several CDDP concentrations. In order to perform this assay and calculate the combination index, we would need at least 6 different CDDP tested concentrations and a fixed concentration of ME and TAP extracts. In our paper, our aim was not to demonstrate the combination index, but rather evaluate whether their effects could sum up and better induce inhibition on cell viability. In none of the tested CDDP concentrations we found a benefit on using CDDP in combination with ME and TAP extract. These data further reinforce that the use of these ME and TAP extracts could be rather tested in a treatment design as previous treatment to CDDP.

8 -How about the cytotoxicity of the fruit extracts on normal ovarian cells?

We kindly thank the reviewer for this suggestion, but these experiments have not been performed and currently we do not have available ovarian non-tumoral cell lines. It can be done in the next step of this investigation project.

Reviewer 2 Report

This manuscript proposes new therapeutic strategies using Amazonian fruit extracts (Murici and tapereba) to sensitize ovarian cancer cells to current chemotherapeutic options. In my point of view the research has been well thought out, but unfortunately lacks in different aspects. Specifically:

In the introduction the authors should added some information about the employed Amazonian fruits. In particular, the data regarding diffusion area were omitted as well as the market information. Please add this information in the section. Results and discussion section should reports on the extraction procedure. How the extraction procedure was optimized? Please justify or insert a convenient reference. Did the authors test different conditions? Antioxidant properties of the extracts can be considered a consequence of different compounds, not only carotenoids. In my opinion, a more complete characterization of the extracts, including quantitative estimation of others polyphenol molecules, should be provided.

According with my point of view, this article is suitable for publication in Biomolecules only after mayor revision.

Author Response

This manuscript proposes new therapeutic strategies using Amazonian fruit extracts (Murici and tapereba) to sensitize ovarian cancer cells to current chemotherapeutic options. In my point of view the research has been well thought out, but unfortunately lacks in different aspects. Specifically:

1- In the introduction the authors should added some information about the employed Amazonian fruits.

A: Thank you for your comments. To meet this expectation we have added an additional paragraph in the introduction, focusing on the aspects of importance, use, production and economy, as below:

“These fruits are endemic in some ecosystems in Brazil but produced predominantly in the Amazon region in forest yards or are then harvested from extractivism. They present an increasing economic importance in tropical regions, especially in the field of eccentric juices'  and desert market [12,13,16].The yield of these fruits according to the Brazilian yearbook of geography and statistics is about 7.8 tons / year in 2018. This activity contributes greatly to the increase of family farmers' income and plays a key role in the local economy by accounting for more than $ 2 million / year [41]”.

2 - In particular, the data regarding diffusion area were omitted as well as the market information. Please add this information in the section.

A: We add an additional paragraph in the introduction responding to these questions

3 - Results and discussion section should reports on the extraction procedure. How the extraction procedure was optimized? Please justify or insert a convenient reference. Did the authors test different conditions?

A: Yes we did.  We tested different types of extraction until we found what we are emphasizing in this article. We added the reference in the section as requested.

4 - Antioxidant properties of the extracts can be considered a consequence of different compounds, not only carotenoids. In my opinion, a more complete characterization of the extracts, including quantitative estimation of others polyphenol molecules, should be provided.

A: In other studies, approximately nineteen polyphenolic compounds have been identified in murici, including galotanines, quinic acid gallates, proanthrocyanidenes, quercetin derivatives, and gallium derivatives, substances rarely found in fruits [10,18]. Authors have found strong correlations between total phenolic compounds and antioxidant activity in fruits, including murici [17]. They have already identified saponins, tannins, flavones, flavonoids, leucoanthocyanidins, alkaloids and glycosides in tapereba pulps [14].

5 - According with my point of view, this article is suitable for publication in Biomolecules only after mayor revision.

A: We were grateful and we received comments in an attempt to get our article ready for publication.  Thank you very much.

Round 2

Reviewer 1 Report

The revised manuscript “Amazon fruits inhibit growth and promote proapoptotic effects on human ovarian carcinoma cell lines” have adequately addressed my previous concerns and the paper is now acceptable for publication.

Reviewer 2 Report

The authors performed the requested modifications and the article is now suitable for publication on Biomolecules.

This manuscript is a resubmission of an earlier submission. The following is a list of the peer review reports and author responses from that submission.

Round 1

Reviewer 1 Report

Thank you for the possibility to review this very interesting manuscript. It covers a curious aspect with potential application character, especially for biochemistry and pharmacology oriented reader. Abstract and Introduction  are well written, all elements are clearly presented. The methodology is well and clearly described. Also discussion with results provides a concise and clear description of a core idea of the manuscript with confrontation of literature of the field. Authors also broadly presented possible practical applications of the study, what is undoubtedly a big strength of the manuscript. In addition, the authors show that this is the first study on these ovarian cell lines.

Nevertheless, I have some comments to be addressed:

In introduction, authors wrote about two species but they don’t write what is the family name of these species, please add in introduction.

In manuscript, in line 32 is l but should be L

What is the viability of normal cells after treatment of plant extracts? Did the authors do such  study?

Keywords should be searchable by the exact word/phrase in the MESH library (https://meshb.nlm.nih.gov/search) it will improve visibility of the study in the internet and will be of benefit for future citations these keywords are too simple-please correct.

Please correct the references because they are written differently e.g line 493,495,500…………

In figure 4 the authors write about apoptosis  percentage? But in the table is percentage of what??? Early apoptosis or late or early +late? I suggest  adding the percentage of apoptosis in the upper corners of each cytogram.

please correct typos and dots in the text  eg. 260 line?

The next time I suggest   authors to make extract determinations by HPLC.

Author Response

Thank you for the possibility to review this very interesting manuscript. It covers a curious aspect with potential application character, especially for biochemistry and pharmacology oriented reader. Abstract and Introduction  are well written, all elements are clearly presented. The methodology is well and clearly described. Also discussion with results provides a concise and clear description of a core idea of the manuscript with confrontation of literature of the field. Authors also broadly presented possible practical applications of the study, what is undoubtedly a big strength of the manuscript. In addition, the authors show that this is the first study on these ovarian cell lines.

R: We thank the reviewer for this positive comment.

 In introduction, authors wrote about two species but they don’t write what is the family name of these species, please add in introduction.

R: We add the family name in introduction section.

 In manuscript, in line 32 is l but should be L

R: We modified keywords according reviewer suggests.

What is the viability of normal cells after treatment of plant extracts? Did the authors do such  study?

R: We thank the reviewers for this comment and understand your point. Unfortunately, we still haven’t tested ovarian non-tumoral cell lines with these same cell extracts regarding the viability assays, once we do not have these cells available in our lab. Added, ovarian non-tumoral cell lines is rare.

Keywords should be search able by the exact word/phrase in the MESH library (https://meshb.nlm.nih.gov/search) it will improve visibility of the study in the internet and will be of benefit for future citations these keywords are too simple-please correct.

R: We modified keywords according reviewer suggests.

 Please correct the references because they are written differently e.g line 493,495,500…………

R: We modified the reference according to MDPI author´s guide instructions.

In Figure 4 the authors write about apoptosis  percentage? But in the table is percentage of what??? Early apoptosis or late or early +late? I suggest  adding the percentage of apoptosis in the upper corners of each cytogram.

R: It is completely correct. The results describe the fold increase rate, so, we excluded the percentage of the text and modified the paragraph. We consider both types (later and early), but it is possible to see in the figure that the late one had more influence.

 Please correct typos and dots in the text  eg. 260 line?

R: We correct typos, dots and English language in text.

The next time I suggest authors to make extract determinations by HPLC.

R: We appreciate the suggestion for a better characterization of the extracts, but due to the large number of standards required, we only characterize the carotenoids by HPLC.

Reviewer 2 Report

The manuscript titled “Amazon fruits inhibit growth and promote proapoptoti c effects on human ovarian carcinoma cell lines” by Rosse de-Sonza etc  described the extraction, characterization  and evaluation of antioxidant  properties of phytochemicals from Murici and tapereba.  Furthermore, their cytotoxic  effect, modulation of cell cycle and induction of apoptosis  were studied in cisplatin-sensitive and resistant ovarian cancer cell lines.  It is important to search for novel therapies against ovarian cancer, in particular the cisplatin resistant ovarian cancer. In this study, some interesting results were obtained but only when very high concentration 5-10 mg/ml of the plant extracts were used in most of the experiments. It is unlikely to achieve such high concentrations in in vivo. There also seems to be no obvious advantages of using these extracts against the cisplatin resistant cells.

The following issues need to be sorted out:

Major:

Positive control- cisplatin is missing in both cell lines How were the six carotenoids identified? E.g., by mass spectrometry or NMR? Figure 4: What do those values in the bottom table represent? If they are the percentage of apoptotic cells, should they be greater based on the top scatter plots? Why is there almost no apoptosis effect in the ACRP cells for both ME and TAP while effective in the MTT assay and cell cycle experiments? Figure 5: There is no analysis to see if there is any synergistic effect in the combination studies. Why 24h rather than 48 or 72 hour was chosen for all the experiments? Plant material need be authenticated and specimen stored.

Minor:

Abstract must be improved. ACRP needs to be clarified as cisplatin resistant cell line; line 28-30, there should be four values for two extract against two cell lines; the concentration is missing. Line 54-57 should be together with the paragraph (line 67-76) Line 195-197, the statement is correct? Table 1 and 2, what do superscript a and b represent? Figure 1 and 2, scale bars are missing, there are no description and discussion about the images provided. Figure 3A: fonts are too small to see. Last paragraph in the Conclusion section should be moved to the discussion section.

Author Response

The manuscript titled “Amazon fruits inhibit growth and promote proapoptotic effects on human ovarian carcinoma cell lines” by Rosse de-Sonza etc  described the extraction, characterization  and evaluation of antioxidant  properties of phytochemicals from Murici and tapereba.  Furthermore, their cytotoxic  effect, modulation of cell cycle and induction of apoptosis  were studied in cisplatin-sensitive and resistant ovarian cancer cell lines.  It is important to search for novel therapies against ovarian cancer, in particular the cisplatin resistant ovarian cancer.

R: We thank the reviewer for this positive comment.

In this study, some interesting results were obtained but only when very high concentration 5-10 mg/ml of the plant extracts were used in most of the experiments. It is unlikely to achieve such high concentrations in vivo. There also seems to be no obvious advantages of using these extracts against the cisplatin resistant cells.

R: We kindly acknowledge your comment. Actually, based on our data, we propose that Taberebá and Murici fruit extracts could be more preventive approches than effective treatment options for ovarian cancer.In these context, we highlight those data obtained when using combined treatment of these fruit extracts with CDDP. These data show that the combined treatment promote a higher decrease on ACRP cell viability when compared with ACRP cell line only treated with murici or taberepá fruit extracts. Then , we suggest that these fruit extracts could be better proposed as ovarian cancer preventive strategies or preventing CDDP resistance or even as treatment options when used in combination with CDDP therapy, sensitizing resistant cells to CDDP compound.

Positive control- cisplatin is missing in both cell lines.

R: Reviewer is absolutely correct when mentioning this limitation. We did not include this positive control, but otherwise we used as a negative control those cells that have not been treated with CDDP in order to compared cell viability of those CDDP treated cells with this control, which has been considered as 100% of cell viability.

How were the six carotenoids identified? E.g., by mass spectrometry or NMR?

R:  We appreciate the suggestion for a better characterization of the extracts, but due to the large number of standards required, we only characterize the carotenoids by HPLC (Lutein; Zeaxanthin; Zeinoxanthin, β-cryptoxanthin; α-carotene and β-carotene).

Figure 4: What do those values in the bottom table represent? If they are the percentage of apoptotic cells, should they be greater based on the top scatter plots? Why is there almost no apoptosis effect in the ACRP cells for both ME and TAP while effective in the MTT assay and cell cycle experiments?

R: Your comments are  completely correct. The results describe the fold increase rate, so, we excluded the percentage of the text and modified the paragraph. We suggest that the cytotoxicity effect by a mechanism other than apoptosis, with extracts capable of decreasing viability more than cell cycle and loss of apoptosis. Other studies can be performed to verify this hypothesis, as well as to better describe the mechanisms of action.

Figure 5: There is no analysis to see if there is any synergistic effect in the combination studies. Why 24h rather than 48 or 72 hour was chosen for all the experiments? Plant material need be authenticated and specimen stored.

R: Sure. The synergistic effect and other time incubation is the next step of the study, but the relationship between the murici and tapereba fruits with ovarian cancer and resistance to chemotherapy is not well described in the literature. In this work, we used fruits that were supplied by a pulp processor from an ecoregional actor that is authenticated and specimen stored.

Abstract must be improved.

R: We improved and modified the abstract.

ACRP needs to be clarified as cisplatin resistant cell line;

R: For the present study, we used the CDDP chemotherapy resistance cell line model used in the ovarian cancer clinic. The ACRP ovarian cancer strain was derived from the A2780 parent strain and is CDDP resistant and has also been generated through exposure to increasing drug concentrations. To maintain the resistance phenotype, treatment was performed 3 consecutive times with 5 μM CDDP for 24 hours [38]. We added this information in method section.

Line 28-30, there should be four values for two extract against two cell lines; the concentration is missing.

R: We improved and modified the abstract.

Line 54-57 should be together with the paragraph (line 67-76)

R:  We modified according reviewer suggestion.

Line 195-197, the statement is correct?

R: We modified the sentence.

Table 1 and 2, what do superscript a and b represent?

R: Different letters on the same line indicate significant difference (p< 0.05). We introduced in table legend.

Figure 1 and 2, scale bars are missing, there are no description and discussion about the images provided.

R: We added the scale in figure legend and provide discussion in the text.

Figure 3A: fonts are too small to see.

R: We improved all fonts in Figure 3.

Last paragraph in the Conclusion section should be moved to the discussion section.

R: We modified according your suggestion.

Round 2

Reviewer 1 Report

The authors  corrected the manuscript according to my suggestions.

Reviewer 2 Report

The authors have improved the revised manuscript,  but unfortunately the key issues still remain to be resolved:

The positive controls using cisplatin are missing;

The accurate identification of compounds using only HPLC is not enough;

There is no detailed analysis of the cisplatin extracts combination study to indicate if there is synergstic, additive or antagonistic effect;

The poor potency of the extracts in both cisplatin sensitive and resistant cells should not be claimed to have therapeutic role in the high impact journal – Biomolecules, which may mislead the future study.